# Use of Maximum Tongue Pressure Values to Examine the Presence of Dysphagia after Extubation and Prevent Aspiration Pneumonia in Elderly Emergency Patients

**DOI:** 10.3390/jcm11216599

**Published:** 2022-11-07

**Authors:** Ryo Ichibayashi, Hideki Sekiya, Kosuke Kaneko, Mitsuru Honda

**Affiliations:** 1Department of Critical Care Center, Toho University Medical Center Omori Hospital, Tokyo 143-8541, Japan; 2Department of Oral Surgery, School of Medicine, Toho University, Tokyo 143-8541, Japan

**Keywords:** post-extubation, aspiration pneumonia, dysphagia, maximum tongue pressure (MTP), oral intake, rehabilitation, elderly emergency patients

## Abstract

Background: Tongue pressure values in patients with dysphagia are reported to be significantly lower than those in healthy controls. The aim of this study was to measure the maximum tongue pressure (MTP) values after extubation in order to assess the presence of post-extubation dysphagia for the safe initiation of oral intake in elderly patients. Methods: Data from 90 patients who were extubated after mechanical ventilation under tracheal intubation were collected retrospectively. The patients were divided into two groups as follows: normal group (those who did not develop aspiration pneumonia after extubation; median age 62 years) and aspiration group (those who developed aspiration during the evaluation period; median age 75 years). The MTP values were measured at 6 h, 24 h, 3 days, and 7 days after extubation. Results: The values were significantly increased 24 h after extubation in the normal group (*p* < 0.05). Alternatively, no increase was observed even after 1 week of extubation in the aspiration group, and the values were significantly lower than those in the normal group. The cutoff values at 6 and 24 h after extubation, which were measured using the receiver operator characteristic (ROC) curve, were 17.8 and 23.2 kpa, respectively; furthermore, the results of these assessments were strongly related to the development of aspiration 6 h after extubation (χ^2^-value: 6.125; *p* = 0.0133). Conclusions: The presence of post-extubation dysphagia in patients who are intubated for ≥24 h can be predicted based on age and the MTP values at 6 h after extubation.

## 1. Introduction

Approximately half of all patients receiving mechanical ventilation with endotracheal intubation experience post-extubation dysphagia [1,2,3]. Older adults and those with underlying diseases might experience pre-existing dysphagia; therefore, the assessment of swallowing function in individuals who have been transported to the emergency department is challenging. Although researchers have speculated that muscle weakness and sensory impairment due to long-term intubation can cause dysphagia after tracheal intubation, no definitive conclusions have been reached so far [4]. Prolonged hospitalization and poor prognosis are common in patients who experience dysphagia after extubation, and could lead to increased medical expenses [5].

Several screening methods, such as the repetitive saliva swallowing test, modified water swallowing test, and other bedside evaluations, can be used to assess tongue movement and the oral environment in order to determine the presence of dysphagia [6]. However, despite the use of these methods, dysphagia can remain undiagnosed in some patients and lead to aspiration or pneumonia. The tongue pressure test is a quantitative measure of the swallowing ability, wherein a numerical value is assigned to assess the tongue function [7,8,9]. Previous studies have indicated that tongue pressure is associated with dysphagia; tongue pressure values in patients with dysphagia were reported to be significantly lower in patients with dysphagia than those in healthy controls, and were correlated with other well-known functional criteria used to evaluate the swallowing ability [9,10]. A maximum value of <30 kPa indicates decreased tongue pressure and might reveal the presence of dysphagia [11]. Tongue pressure measurements are easy to obtain owing to the noninvasive nature of the method used during bedside evaluations of the swallowing function after extubation; additionally, this method yields reproducible quantitative values [12]. The aim of the present study was to measure the maximum tongue pressure (MTP) values in patients who underwent extubation in order to determine whether these values can be used to assess the presence of post-extubation dysphagia for the safe initiation of oral intake of foods.

## 2. Participants and Methods

### 2.1. Participants

The sample size required for this study was calculated assuming a 1:2 ratio of patients with and without post-extubation aspiration findings [5]. Therefore, data from a total of 93 patients were required to obtain an area under the curve (AUC) value calculated from the receiver operating characteristic (ROC) curve of ≥0.7 (power of detection 95%). Subsequently, data from 90 patients who were brought to the Emergency Life Support Center at our hospital from 2012 to 2016 were collected. Those who received mechanical ventilation with endotracheal intubation for 24 h or longer, and whose primary condition was successfully treated resulting in extubation, were included in the study. The exclusion criteria were as follows: cases where it was not possible to obtain regular measurements, patients who underwent re-intubation or experienced delirium within 7 days after extubation, and patients with neurologic/cerebrovascular disorders. All participants were evaluated by using the repetitive saliva swallowing test [13] and modified water swallowing test [6], and participants without dysphagia were initiated on oral intake. The participants without dysphagia resumed oral intake with the same form of food as before their ICU admission and swallowing rehabilitation from post-extubation to the start of oral intake consisted of indirect training as the usual protocol used in our hospital.

The present study was approved by the Toho University Medical Center Omori Hospital Ethics Committee. Written explanations regarding the aims and procedures used in the study were provided, and informed consent was obtained from all the participants or their family members prior to enrollment (Approval No. 24-132).

### 2.2. Methods and Data Collection

A TPM-01 device (JMS, Hiroshima, Japan) was used to measure the tongue pressure (Figure 1). The MTP values were measured at 6 h, 24 h, 3 days, and 7 days after extubation. The maximum values of three individual measurements collected at each time point were recorded in kPa. The actual measurements were obtained by a designated nurse at the Emergency and Critical Care Center, or by members of a specially trained swallowing support team in the regular wards. All evaluators received specific instructions regarding the tongue pressure measurements to ensure inter-observer reliability.

The characteristics of the patients, including age, sex, tracheal intubation period, Acute Physiology and Chronic Health Evaluation II (APACHE II) score, and causative disease, were retrospectively evaluated. The patients were divided into two groups as follows: those who did not develop aspiration pneumonia after extubation (normal group) and those who developed aspiration during the evaluation period, resulting in aspiration pneumonia or signs of aspiration pneumonia with the cessation of oral intake (aspiration group). Newly or recurrent visible infiltrative shadows on recent radiographs obtained from the medical records were required for signs of aspiration pneumonia. In addition, the ability to use the MTP values obtained at 6 and 24 h after extubation to assess dysphagia in order to predict the optimum time for the safe initiation of oral intake in the normal and aspiration groups was examined. These values were chosen owing to the retrospective nature of this study, in order to adjust for the background characteristics before starting the oral intake. Oral intake was initiated 24 h after extubation at the discretion of the attending physician, but the MTP values were not used to determine whether the patients were ready to start the oral intake at that time.

Since the primary endpoint of this study is the prediction of pneumonia at 24 h after extubation and after initiation of oral intake, data on MTP value at days 3 and 7 are presented as a secondary endpoint to the results.

### 2.3. Statistical Analysis

EZR ver1.52 (Kanda, 2014, Saitama, Japan [15]) was used for statistical analysis. The Mann–Whitney U-test was used to analyze differences in age, intubation time, and APACHE II scores, whereas the Chi-square test was used to examine differences in the male-female ratio and the causative diseases. Associations with MTP values after extubation were analyzed using the Friedman’s test or Dunn’s multiple comparison test. The MTP values at 6 h, 24 h, 3 days, and 7 days after extubation were compared between the normal and aspiration groups using the Mann–Whitney U-test. In addition, ROC curve analysis was performed to examine the MTP values at 6 and 24 h after extubation, in order to determine whether these values can be used to predict aspiration. The AUC values and the diagnostic cutoff value for MTP at 6 and 24 h were calculated from the ROC curves using Youden’s J statistic. Statistical analyses of the ROC curve were performed also using EZR ver1.52.

Subsequently, a multivariable logistic regression analysis was performed to calculate the result. The objective variable for the analysis was the detection of aspiration pneumonia, and the explanatory variables were “Age above 75”, “Causative diseases”, “Low MTP value (6 h)”, and “Low MTP value (24 h)”. Data pertaining to age were categorized into groups (those younger than 75 years and above 75), because the age of the normal group was 75.6 years which is 1 SD more than the mean age. The median age of the aspiration group was 75 years, and this is due to the fact that the elderly classification in Japan is 75 years old or older. Causative diseases of intubation were classified into two groups. One group consisted of pneumonia, sepsis, exacerbation of chronic obstructive pulmonary disease (COPD), and acute pancreatitis, which cause ARDS; the other group consisted of other diseases. The MTP values at 6 and 24 h were classified as “low” and “high”, respectively, as determined by their cutoff values. In addition, an ROC curve analysis was performed on the combined “Age above 63”, “Low MTP value (6 h)”, and “Low MTP value (24 h)”. A two-tailed *p*-value of <0.05 was considered statistically significant. The multivariable logistic regression analysis was also performed using EZR ver1.52.

The independent models were adjusted by confounders for the different times of the MTP using pair-matching analysis. We assumed age and causative disease as confounding factors. After correcting the confounding factors, the Mantel–Haenszel Chi-squared test with continuity correction was performed to examine the possibility of predicting aspiration at each time-point using EZR ver1.52.

## 3. Results

The normal and aspiration groups comprised 70 and 20 patients (22.2%), respectively; the background characteristics of the patients in the two groups are shown in Table 1. No significant differences in sex, endotracheal intubation period, or severity were observed between the groups. However, patients in the aspiration group were significantly older than those in the normal group (*p* < 0.05). Furthermore, no significant differences in the causative diseases for intubation were observed between the two groups (Table 2).

Changes in MTP values after extubation in the entire study population and in the two groups are shown in Figure 2. The MTP values were significantly (*p* < 0.05) increased to 20.7 kPa, 21.6 kPa, 24.9 kPa, and 26.7 kPa at 6 h, 24 h, 3 days, and 7 days after extubation, respectively, over a period of 1 week in the entire study population (Figure 2A). Similar findings were observed in the normal group (25.5 kPa, 24.2 kPa, 28.1 kPa, and 30.3 kPa at 6 h, 24 h, 3 days, and 7 days after extubation, respectively; Figure 2B). On the contrary, no significant increase was observed in the aspiration group, which exhibited significantly lower MTP values than those in the normal group (13.9 kPa, 18.7 kPa, 11.1 kPa, and 13.1 kPa at 6 h, 24 h, 3 days, and 7 days after extubation, respectively; Figure 2C). Significant differences in MTP values were observed between the groups at each time point (Figure 3). Multiple comparison tests revealed significant differences between the MTP values at 6 h and 3 days, 6 h and 7 days, 24 h and 3 days, and 24 h and 7 days (*p* < 0.05). No significant differences were observed between the 6- and 24-h mark (*p* = 0.83).

ROC curve analysis was performed to determine whether the MTP values after 6 and 24 h could be used to predict dysphagia. The diagnostic cutoff MTP value after 6 h was 17.8 kPa, the sensitivity was 80.0%, the specificity was 67.1%, the positive predictive value was 41.0%, and the negative predictive value was 92.2% (Figure 4A). The corresponding values (diagnostic cutoff, sensitivity, specificity, positive predictive, and negative predictive) after 24 h were 23.2 kPa, 90.0%, 52.9%, 35.3%, and 94.9%, respectively (Figure 4B).

The AUC values were calculated from the ROC curves using Youden’s J statistic. The values after 6 h and 24 h were 0.79 (95% CI: 0.69–0.89) and 0.74 (95% CI: 0.61–0.86), respectively.

Multivariable logistic regression analysis was performed to determine the factors associated with aspiration after extubation (Table 3). “Age above 75” and “Low MTP value (6 h)” were the significant variables; therefore, ROC curve analysis was performed to investigate the predictive ability of the combination of “Age above 75”, “Causative diseases”, “Low MTP value (6 h)”, and “Low MTP value (24 h)” for dysphagia (Figure 5). The AUC value was 0.81 (95% CI: 0.73–0.90).

We hypothesized that the combination of these items would aid in predicting post-extubation dysphagia, and “Age above 75” and “Causative diseases” were considered confounding factors. Therefore, a model excluding the confounding factors was created using the pair-matching analysis. A total of 40 cases, 20 in each of the two groups, was selected for the analysis (Table 4). The Mantel–Haenszel Chi-squared test was performed to examine the possibility of predicting aspiration at the 6 and 24 h time points, as shown in Table 5.

## 4. Discussion

The background factors were related to age in patients belonging to the normal and aspiration groups in this study; those in the aspiration group tended to be older than those in the normal group. The initiation of oral intake in elderly patients after life support should be approached with caution. Therefore, it is important to design an evaluation method that can be used to determine the optimum time for oral intake after extubation. No clear relationship was noted for either pneumonia or ARDS causal disease as the causative disease for intubation, which was speculated to be possibly related to post-extubation pneumonia. Prolonged endotracheal intubation is a common iatrogenic cause of swallowing disorders, but as shown in Table 1, there was no additional difference between the two groups (*p* = 0.50), so it was not included as a variable in the multivariate analysis. This study included patients with severe disease (APACHE II: around 24), which is the reason for the higher incidence of pneumonia compared to previous reports [1,2,3].

The aim of the present study was to measure the MTP values in patients who underwent extubation in order to determine whether they can be used to assess the presence of post-extubation dysphagia, which could affect the initiation of oral intake in older patients. The participants in this study had normal RSST and modified water swallowing test results, so the development of post-extubation pneumonia or aspiration would be unlikely if true, but in 20 of the 90 cases, this occurred. This study suggests that post-extubation pneumonia due to dysphagia, which cannot be detected by conventional screening methods, may be predicted by using maximal tongue pressure in combination with conventional methods. Our findings indicated that the MTP values were initially low in patients requiring mechanical ventilation with endotracheal intubation for more than 24 h, but returned to normal levels after 1 week among those without dysphagia. These findings are in accordance with those of our previous study, which showed that patients with dysphagia exhibited lower MTP values, which did not significantly increase even after 1 week [14]. Evidence suggests that contact with the endotracheal tube during long-term oral intubation causes muscle weakness in the tongue and sensory impairment, leading to dysphagia; however, the underlying mechanisms remain unclear [4]. Early extubation and the use of appropriately-sized endotracheal tubes have been reported to reduce the risk of dysphagia [16]. However, other studies have reported no associations between the endotracheal intubation period and dysphagia [17].

Although no significant differences in MTP values were observed at 6 and 24 h after extubation in the normal group, the values were significantly increased after 3 days. It remains unclear as to why the MTP values did not recover within the first 24 h. Strict respiratory monitoring is required for at least 24–48 h after extubation [18] because hoarseness, laryngeal edema, and an increase in intraoral and sputum discharge caused by extubation can increase the risk of re-intubation during this period. Furthermore, the breathing effort is reported to increase, leading to fatigue of the respiratory muscles 1 h after extubation [19]. Heavy secretion in the oral cavity and increased sputum discharge are thought to delay the recovery of the swallowing function. Weakness of the swallowing muscles due to respiratory fatigue is known to continue for more than 24 h [20], which might account for the absence of any increase in the MTP value within the first 24 h after extubation. These findings highlight the importance of the respiratory status in relation to the swallowing function.

In the aspiration group, the MTP values did not recover 1 week after extubation; the MTP values were significantly lower in the aspiration group (13.9 kPa) than in the normal group (25.5 kPa) 6 h after extubation. Although the values had recovered to 18.7 kPa in the aspiration group 24 h after extubation in this study, they were below the normal values (30 kPa) [8,11]. Thus, it is important to measure the MTP value within 24 h of extubation to predict dysphagia.

The negative predictive values observed in the present study indicated that the ROC curve for MTP values could be used as an index for predicting aspiration 6 and 24 h after extubation. The diagnostic cutoff value of the MTP value after 6 h was 17.8 kPa, and the negative predictive value was 92.2%. The diagnostic cutoff value of the MTP value after 24 h was 23.2 kPa, and the negative predictive value was 94.9%. However, even if the MTP value is low, dysphagia may not necessarily be present. Thus, it may be necessary to evaluate other factors when examining the swallowing function following extubation. MTP values differ based on age and sex [8]. The findings of the current study suggest that age is a risk factor for aspiration after extubation. Tsai et al., reported that patients over 65 years of age are more likely to develop dysphagia after extubation, and should be monitored and treated for 1–2 weeks before resuming oral intake of food [21]. Older adults may exhibit a weak cough reflex or latent aspiration prior to hospitalization, which could increase the incidence of dysphagia due to tracheal intubation. Furthermore, various factors such as cognitive decline, a high incidence of delirium, and a history of taking multiple medications can increase the risk of aspiration in older adults.

Both age and pre-admission information regarding swallowing function are necessary to determine when oral intake can be re-initiated. Although it is difficult to obtain accurate information from the patient in the emergency department, our findings showed that MTP values 6 and 24 h after extubation were strongly associated with aspiration. Thus, when combined with other assessments, these values can be used to determine when oral intake can be initiated safely.

The average time for the re-initiation of oral intake after heart disease surgery is 118.4 h in patients with swallowing disorders [1]. The swallowing reflex can be restored within 1 week in patients who undergo long-term intubation in the intensive care unit (ICU) [22]. The length of hospitalization is extended in patients who develop swallowing disorders after extubation, thereby increasing the cost of treatment. The MTP values increased within 1 week after extubation in the present study; therefore, clinicians can use these values to determine the efficacy of rehabilitation and the time at which oral intake can be resumed after dysphagia, which could aid in shortening the hospitalization period and decreasing the treatment costs.

The results of this study indicate that 24 h after extubation, one can determine whether oral intake can be resumed. Oral intake can be safely initiated under the following conditions: in patients <75 years of age, when the MTP value is >17.8 kPa at 6 h after extubation, and the MTP value does not decrease after 24 h (remains > 23.2 kPa). In older emergency patients who do not meet these conditions, additional swallowing screening tests, videoendoscopy (VE) or videofluorography (VF), should be performed to diagnose the presence of dysphagia. Furthermore, the findings of this study suggest that regular measurement of the MTP is useful to determine the effectiveness of the rehabilitation and the appropriate time for resumption of oral intake.

In future studies, we need to design an interventional study in which patients with anticipated dysphagia determined by a new protocol with tongue pressure, were given rehabilitation prior to oral intake. We will then examine if the incidence of post-extubation pneumonia would decrease from 22%.

The MTP values may be of adjunctive use for other lung diseases, such as interstitial pneumonia with repeated acute conversions and dysphagia due to sarcopenia after COVID-19 infection [23]. Clinicians may find it challenging to determine the optimum time for the resumption of oral intake in patients with these diseases; thus, the use an ancillary tool other than the conventional swallowing assessment tests might prove beneficial.

## 5. Conclusions

This study demonstrated that the presence of post-extubation dysphagia can be assessed based on the MTP values at 6 and 24 h after extubation. This information can be used to determine the time for the initiation of oral intake in older emergency patients who have been intubated for ≥24 h. Additionally, changes in MTP values may be used to determine the optimum time when oral intake can be resumed and the efficacy of the rehabilitation following dysphagia. However, MTP values should be used in combination with other swallowing assessments to determine the most appropriate time for re-initiating oral intake. Additional study will be required to determine the most appropriate swallowing assessment that can be used in conjunction with MTP in the future.

## Figures and Tables

**Figure 1 jcm-11-06599-f001:**
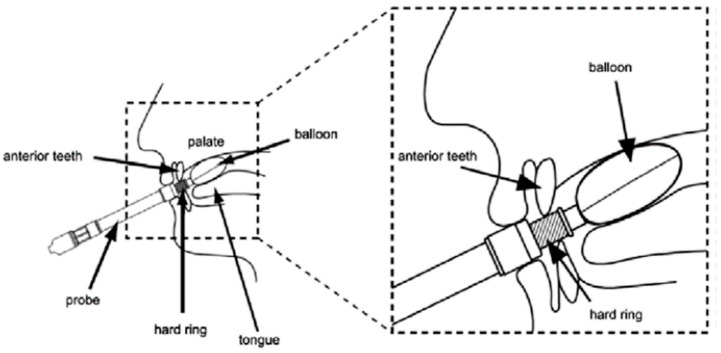
The method used to measure the tongue pressure [14]. The internal pressure of the balloon was adjusted to a predetermined pressure, after which it was positioned in the oral cavity as shown. The tongue pressure was measured by asking the patient to lift the tongue toward the palate with as much force as possible, as if to crush the balloon.

**Figure 2 jcm-11-06599-f002:**
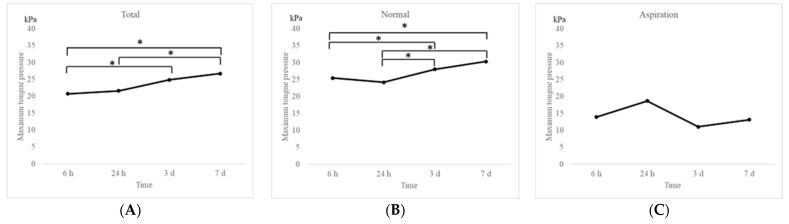
Graphs showing changes in MTP in the entire study population (**A**) and in the two groups (**B**,**C**). MTP values recovered over a period of 1 week after extubation in the entire study population (**A**) and the participants in the normal group ((**B**); Friedman test, *p* < 0.05). (**C**) The MTP values remained low even after 1 week in the aspiration group. * *p* < 0.05; h: hours; d: days. Friedman’s test, Dunn’s test.

**Figure 3 jcm-11-06599-f003:**
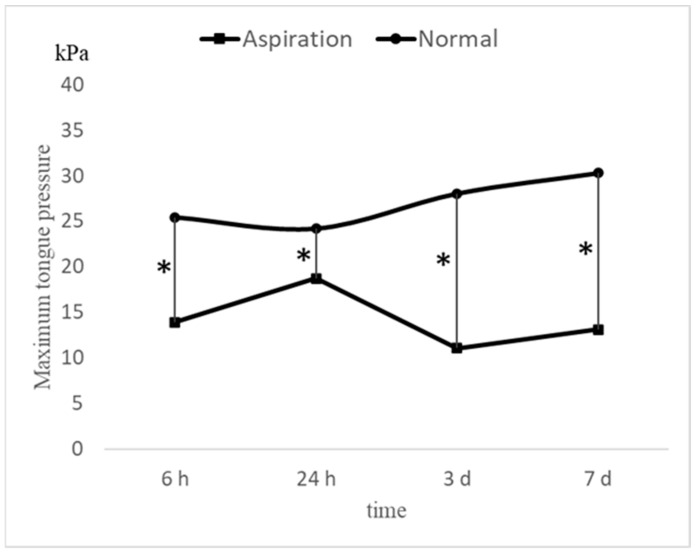
MTP values in the normal and aspiration groups. * *p* < 0.05; h: hours; d: days. Mann–Whitney U-test.

**Figure 4 jcm-11-06599-f004:**
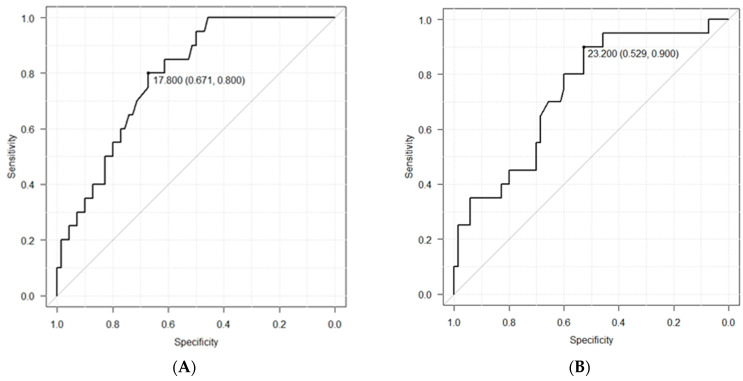
Receiver operating characteristic (ROC) curve analysis 6 and 24 h after extubation. (**A**) ROC curve analysis 6 h after extubation. (**B**) ROC curve analysis 24 h after extubation. It is possible to estimate diagnostic cutoff values from ROC curves.

**Figure 5 jcm-11-06599-f005:**
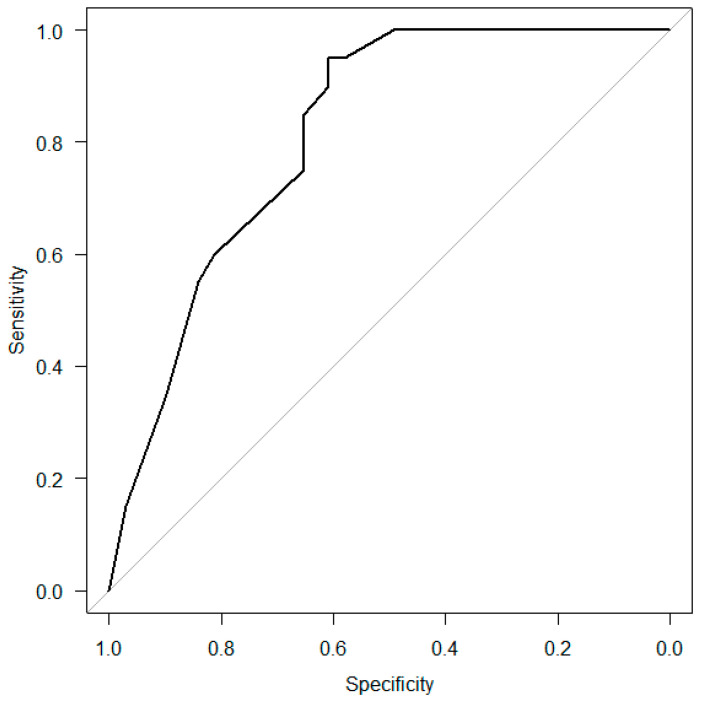
ROC curve analysis on the combined variables “Age above 75”, “Causative diseases”, “Low MTP value (6 h)”, and “Low MTP value (24 h)”. The AUC value was 0.81 (95% CI: 0.73–0.90).

**Table 1 jcm-11-06599-t001:** Background characteristics of the patients in the normal and aspiration groups.

	Total (*n* = 90)	Normal (*n* = 70)	Aspiration (*n* = 20)	*p*-Value
Age (years); Average ± SD	62.9 ± 15.8	59.7 ± 15.9	73.9 ± 9.5	<0.01
Median	66	62	75	
Sex (male/female)	55/35	42/28	13/7	0.69
Intubation period (days)				0.5
Average ± SD	8.1 ± 5.5	8.0 ± 5.8	8.5 ± 4.2	
Median	7	6.5	9	
MTP value (at 6 h)				<0.01
Average ± SD	23.0 ± 10.1	24.8 ± 11.6	13.2 ± 7.5	
Median	21.6	25.5	13.9	
MTP value (at 24 h)				<0.01
Average ± SD	25.3 ± 11.7	24.9 ± 9.7	16.4 ± 8.8	
Median	24.9	24.2	18.7	
APACHE II score				0.81
Average ± SD	23.9 ± 9.8	23.6 ± 10.0	24.8 ± 9.3	
Median	23	22.5	23	

SD, standard deviation; APACHE II, Acute Physiology and Chronic Health Evaluation II.

**Table 2 jcm-11-06599-t002:** The disease background of the patients in the two groups.

	Normal (*n* = 70)	Aspiration (*n* = 20)	*p*-Value
CPA recovery	10	2	0.47
Trauma	5	1	0.60
Acute myocardial infarction	7	3	0.39
Heart failure	4	2	0.40
Pneumonia	13	6	0.21
Sepsis	7	4	0.20
Acute pancreatitis	2	0	0.44
COPD exacerbation	1	0	0.59
Others	21	2	0.07

CPA, cardiopulmonary arrest.

**Table 3 jcm-11-06599-t003:** Factors associated with aspiration after extubation by logistic regression analysis.

	Odds Ratio (95% CI)	*p*-Value
Age above 75	3.24(1.01–10.4)	**0.048**
Causative diseases	0.472 (0.14–1.54)	0.214
Low MTP value (6 h)	4.25 (1.02–17.7)	**0.047**
Low MTP value (24 h)	3.26 (0.55–19.3)	0.194

CI: confidence interval. Bold: *p*-values indicate significance.

**Table 4 jcm-11-06599-t004:** A new population excluding the confounding factors was created using the pair-matching analysis. A total of 40 cases, 20 in each in of the two groups, was selected for the analysis.

Age	Normal	Aspiration
<75 years	9	9
≥75 years	11	11
**Causative Diseases**	**Normal**	**Aspiration**
Pneumonia + ARDS Causal Diseases	9	9
Other	11	11

**Table 5 jcm-11-06599-t005:** The Mantel–Haenszel Chi-squared test with continuity correction was performed on each time point after correcting the confounding factors.

	χ^2^-Value	*p*-Value
MTP value (at 6 h)	6.125	**0.0133**
MTP value (at 24 h)	2.50	0.1138

Bold: *p*-values indicate significance.

## Data Availability

The data presented in this study are available on request from the corresponding author. The data are not publicly available due to privacy and ethical considerations.

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
