# Peer review of "Use of Maximum Tongue Pressure Values to Examine the Presence of Dysphagia after Extubation and Prevent Aspiration Pneumonia in Elderly Emergency Patients"

_jcm, 2022, doi:10.3390/jcm11216599_

Round 1

Reviewer 1 Report (Previous Reviewer 2)

I thank the authors for addressing my comments.

Author Response

Dear Reviewer 1,

We are the ones who should be thanking you. You have revised this paper and made it better, and we really appreciate you accepting it.

In addition, due to the short revision deadline of 5 days, we will ask MDPI's English service to check the English text by native speakers after this revision.

Thank you again.

Reviewer 2 Report (New Reviewer)

This paper is really innovative, clinically possible and important. However, some considerations must be answered:

1.                In the method, the authors present that "the diet was only released under medical criteria. The value of tongue pressure was not considered to determine whether or not patients could start oral diet." Why not?

2.                There is no explanation about which criteria were used for diet evolution (validated protocol) - selection of bolus type (e.g., consistency, volume) for each trial;

3.                There is no explanation about the cases in which it was necessary to adapt consistency to safe swallowing;

4.                That is, they released diet for a group, but it was not explained whether these patients had dysphagia. Furthermore, in the discussion and conclusion it is written "the objective was to determine whether the language pressure measure can be used to determine dysphagia". Then, again, they associated dysphagia ONLY with the presence or absence of aspiration pneumonia. Perhaps a swallowing classification scale could explain this point;

5.                Prolonged endotracheal intubation is a common iatrogenic cause of swallowing disorders, there is no explanation about that important variable;

6.                Another point mentioned in the results was: "patients who presented aspiration did not have improvement in tongue strength after 7 days". However, it was not clarified whether these patients underwent swallowing rehabilitation. Also in the results, the  percentage of 22% of aspiration pneumonia in extubated is really high. It is different from epidemiological data.

Author Response

Dear Reviewer 2,

We would like to thank you for your time and effort to review this manuscript.  Please consider the attached manuscript that has been revised according to the reviews.  Response to your comments in attached file. Thank you.

This manuscript is a resubmission of an earlier submission. The following is a list of the peer review reports and author responses from that submission.

Round 1

Reviewer 1 Report

The authors aimed to determine whether MTP can be used to predict the optimum timing for the safe initiation of the oral intake of foods. I would like to congratulate the authors for their effort; however, after reviewing the manuscript, several concerns have arisen that make me consider the manuscript unsuitable for publication.

Major comments:

1. As the authors reported, in all patients, oral intake was initiated 24 h after extubation at the discretion of the attending physician. Hence, in my understanding, the authors did not assess "the optimum timing" outcome, but the dysphagia (yes/no) or tolerability at 24hrs.

2. Leaving aside the appropriate terminology for the outcome, the main objective of the study would be answered by the construction of its AUC ROC. However, they do not present the AUC value (and its 95% CI). They only reported the cut-off value (without mentioning the method [e.g. youden, liu, nearest]) and its Se, Sp, PPV, NPV. This is insufficient to answer the research question. In addition, the authors only based on the NPV for concluding that the MTP can be used to predict the optimum timing for the same initiation of the oral intake of foods. This is clearly not correct. The authors should: 1) present the sample size (or power calculation) for answering their research question; 2) present the AUC value and its 95%CI; 3) describe the method used for the calculation of the cut-off point, although I'm not sure if it would be worth doing additional analysis if the AUC is less than 0.7. In fact, there is no good Se or Sp; 4) if the AUC is good, the authors could perform a regression analysis (using as exposure the dichotomized variable according to the cut-off point), although I'm not sure if they would have the adequate statistical power. 

Minor comments:

1. In table 1, the authors mention "average" and "mean", but mean=average. Is it really the mean or the median?

2. It is not clear what the gap in the literature is. This should be better clarified in the introduction.

In summary, I believe that the methodology, analysis, and results presented in the study are not adequate for assessing the authors' primary objective and can't provide a correct answer to their research question.

Author Response

Dear Reviewer 1,

We sincerely appreciate your detailed and helpful guidance.  We have added all the points you indicated to the best of our ability.

Reviewer 2 Report

In the study by Ichibayashi et al., the authors determine the optimum timing for the safe start of taking food via mouth after extubation. The authors predicted the timings by measuring the maximum tongue pressure (MTP) in patients who underwent extubation. A device measuring the tongue pressure has a balloon placed in the oral cavity. They studied the included study participants into two groups; the Aspiration group and the Normal.

It was predicted that safe initiation of oral intake is equal to or more than 24 hours of extubation.

Overall, the data appear to be diligently obtained, are transparently described, and are an important contribution to the clinics. The discussion could be expanded into more general considerations about the other lung disease and COVID-19.

Author Response

Dear Reviewer 2,

Thank you for the positive feedback regarding this paper. As you indicated, we have added a description of dysphagia caused by respiratory disease and disuse due to Covid-19 in the Discussion. Please, we would like to publish this paper. Thank you in advance for your cooperation.

Sincerely,

Hideki Sekiya

Round 2

Reviewer 1 Report

The manuscript has improved. However, there are still some gaps:

1. The title still does not reflect the real aim of the study. In my understanding, the authors did not examine the optimum time for the safe initiation of oral intake after extubation. It seems to me that they assessed the prognostic performance of the MTP for the development of dysphagia, but I can't be sure because, as I mention, it is not clear to me what is the true aim of the study.

2. Could you check again the sample size calculation? I have made the calculation in the easyROC online software and obtained different values with the parameters you mention: power: 0.95, type 1 error: 0.05, AU: 0.7, allocation ratio: 2. BTW, the allocation ratio is obtained with the formula N2/N1, which could be no dysphagia / dysphagia (if this is the outcome)

3. It is still not clear what the gap in the literature is (or I cannot see it because there are no tracked changes). This should be better clarified in the introduction.

4. I would suggest independent models for the different times of the MTP. In addition, I wouldn't recommend the stepwise methods. An epidemiological approach would be better (ie. independent models adjusted by confounders, which could be selected using DAGs).

Author Response

Dear Reviewer1,

Thank you for your kind guidance, corrections and advice on this paper.

I believe this paper is better because of your review.

I have included the revisions in the attachment and in the manuscript. Thank you in advance for your kind review.

English sentences were checked by native speakers. Thanks for the advice again.

Sincerely,

Hideki Sekiya
